# Gait Variability Using Waist- and Ankle-Worn Inertial Measurement Units in Healthy Older Adults

**DOI:** 10.3390/s20102858

**Published:** 2020-05-18

**Authors:** Timo Rantalainen, Laura Karavirta, Henrikki Pirkola, Taina Rantanen, Vesa Linnamo

**Affiliations:** 1Faculty of Sport and Health Sciences, Neuromuscular Research Center, University of Jyväskylä, 40014 Jyväskylä, Finland; henrikki.j.pirkola@student.jyu.fi (H.P.); vesa.linnamo@jyu.fi (V.L.); 2Faculty of Sport and Health Sciences and Gerontology Research Center, University of Jyväskylä, 40014 Jyväskylä, Finland; laura.i.karavirta@jyu.fi (L.K.); taina.rantanen@jyu.fi (T.R.)

**Keywords:** wearable, gait, accelerometer, dynamics, non-linear

## Abstract

Gait variability observed in step duration is predictive of impending adverse health outcomes among apparently healthy older adults and could potentially be evaluated using wearable sensors (inertial measurement units, IMU). The purpose of the present study was to establish the reliability and concurrent validity of gait variability and complexity evaluated with a waist and an ankle-worn IMU. Seventeen women (age 74.8 (SD 44) years) and 10 men (73.7 (4.1) years) attended two laboratory measurement sessions a week apart. Their stride duration variability was concurrently evaluated based on a continuous 3 min walk using a force plate and a waist- and an ankle-worn IMU. Their gait complexity (multiscale sample entropy) was evaluated from the waist-worn IMU. The force plate indicated excellent stride duration variability reliability (intra-class correlation coefficient, ICC = 0.90), whereas fair to good reliability (ICC = 0.47 to 0.66) was observed from the IMUs. The IMUs exhibited poor to excellent concurrent validity in stride duration variability compared to the force plate (ICC = 0.22 to 0.93). A good to excellent reliability was observed for gait complexity in most coarseness scales (ICC = 0.60 to 0.82). A reasonable congruence with the force plate-measured stride duration variability was observed on many coarseness scales (correlation coefficient = 0.38 to 0.83). In conclusion, waist-worn IMU entropy estimates may provide a feasible indicator of gait variability among community-dwelling ambulatory older adults.

## 1. Introduction

Gait variability refers to the phenomenon that each step/stride differs slightly from the next one [1]. The quantification of gait variability can be operationalised by measuring the variation in spatiotemporal step-to-step or stride-to-stride gait characteristics or more holistically as gait rhythmicity, complexity or smoothness [1,2]. Gait variability tends to increase with age [1] with worsening cognitive abilities [3,4], and has been shown to predict impending adverse health outcomes among initially healthy older individuals [5,6,7,8,9]. Typically, gait variability is captured in a laboratory environment using force plates or pressure-sensitive walkways; however, there has been a concerted effort to enable capturing gait variability using wearable sensors, such as inertial measurement units (IMU) [10,11,12,13,14,15,16]. Wearable sensors, in addition to the affordability, enable assessments in the habitual environment rather than in a laboratory, potentially improving the ecological validity of gait data while also prolonging the potential for sample duration [5,7,17].

While the practical utility of waist-worn sensors has already been established [5,7,17], the reliability and concurrent validity of wearable sensors for gait variability assessments is still unsure to date. That is, the concurrent validity [10,11,12,13,14,15,16] and reliability of a mean step or stride characteristics assessment using waist-, ankle- and foot-worn IMUs has been convincingly established [10,11,18,19]. On the other hand, only a few studies have investigated the quantification of gait variability measures based on IMUs [20,21]. Riva and colleagues showed that holistic measures, which consider the whole time series of the sampled signal (e.g., multiscale sample entropy) as opposed to discretizing the measurement into particular events (e.g., heel-strikes), required fewer strides to be sampled to produce reliable values within testing session [2]. However, estimates of the day-to-day reliability of IMU-assessed gait variability, whether discretized or holistic, are scarce for community-dwelling older adults.

As explained in detail by Ihlen and colleagues [5], entropy is a measure of complexity, and to capture physiologically interesting phenomena occurring at different temporal scales, entropy can be evaluated using multiple temporal scales. Of the various options that provide a holistic measure of signal dynamics potentially descriptive of gait variability, multiscale sample entropy is one of the most established in that it has been shown to differ between fallers and non-fallers in a cross-sectional setting [22], and to be predictive of prospective falls among community-dwelling older individuals [5,7]. Therefore, the purpose of the present study was to establish the session-to-session reliability of waist-worn IMU-based multiscale sample entropy and ankle-worn IMU-based stride duration variability among older adults 70 years of age and older. In addition, the concurrent validity compared to force plate-derived stride duration variability was evaluated.

## 2. Materials and Methods

Twenty-seven healthy men (*n* = 10) and women (*n* = 17) volunteered to participate in the present study. We have previously reported some results from this dataset [21], but briefly, the inclusion criteria included being aged 70 years of age or older and the ability to walk continuously for three minutes without assistive devices. People with acute or chronic unstable cardio-vascular diseases were excluded. The study was approved by the ethical committee of the University of Jyväskylä (5 April 2018), was conducted in agreement with the Helsinki declaration, and informed written consent was obtained from all participants.

The participants attended two measurement sessions a week apart at the University of Jyväskylä biomechanics laboratory. The protocol was explained to the participant and the participant was prepared for the testing session by asking the participant to wear two inertial measurement units (the 3-dimensional accelerations were ±16 g, the rotations were ±2000°/s and the magnetic field was ±1300 μT recorded at 400 Hz; 400 Hz and 20 Hz sample rates, respectively. NGIMU, x-io Technologies, Bristol, UK). One on the right leg was strapped on with an elastic Velcro belt just above the lateral malleolus, and another on the waist just below the iliac crest in the mid-line of the back around the L4 level was also strapped on with an elastic Velcro belt. After instrumentation, the participants were asked to walk up and back a 14 m track continuously for three minutes at their preferred pace. The central 10 m of the track was covered by a custom-made force platform (total of 16 force platforms arranged in two adjacent rows back to back to form a 2 by 8 array. The surface dimensions were 1.25 × 0.60 m, the natural frequency was 180 ± 10 Hz in the vertical direction and 130 ± 10 Hz in the horizontal direction, the linearity was ≤1%, cross-talk ≤ 2%; Raute Precision, Finland [23]) with the 3-dimensional forces recorded using an analog-to-digital board (Vicon T40, Oxford, UK) at a 1000 Hz sampling rate. The inertial signals and the force signal were recorded concurrently during the continuous 3 min walk. The signals were synchronized with a 1.5 V square pulse applied concurrently to the auxiliary channel of the respective recording devices at the end of the recording. The waist-worn IMU recording failed on the second measurement session on one participant, and therefore *n* = 26 for the waist-worn IMU reliability analyses.

To avoid targeting behaviors, the participants were not asked to walk on any particular portion of the force plates and the signals from the whole 10 m force plate array were recorded as the sum of the three directions from all of the individual force plates. Due to both feet being on the force plate array simultaneously, we did not attempt to divide the stride cycle into stance and swing phases. Rather, the stride durations were identified from the recorded signal with the following approach. Heel-strike instants were first identified (Figure 1). All continuous peaks of at least 2.5 ms above 1.1 times body weight were identified from the vertical force. Then, a 50 ms search window was defined backwards from the timing of the peak value of each identified continuous force peak, and the peak positive jerk (first time derivative of force) within the window was defined as a heel-strike candidate. False heel-strikes were then removed by removing all candidates where the maximum jerk was less than 0.3 times the value of the 50 highest candidate values. The remaining heel-strike candidates were then divided into right and left foot heel-strikes based on the horizontal forces subsequent to the candidate heel-strike instant. Only the right foot heel-strike candidates were considered further. Subsequent heel-strike candidates were considered a stride, and all strides with a duration within 1.25 ratio of the median duration of all candidate strides were included as strides reported in the present study. This approach resulted in identifying all heel-strikes based on visual inspection and led to identifying between 66 and 103 (mean 84.3) right leg strides from the continuous 3 min walking trial. The mean stride duration and standard deviation of the stride durations are reported as the outcomes. The standard deviation of the stride durations was used to indicate the stride variability. The percent coefficient of variation (standard deviation of stride durations divided by the mean stride duration multiplied by 100) is also reported as an indicator of variability.

The approach reported in our previous study from this dataset based on sagittal plane angular velocity was used to identify heel-strike and toe-off instances from the ankle-worn inertial measurement unit [21] (Figure 1). The heel-strike and toe-off instances were merged and sorted and then any heel-strikes not followed by a toe-off or toe-offs not followed by a heel-strike were filtered out. Subsequently, any strides defined as heel-strike to the next heel-strike with a duration not within 1.25 of the median of all identified strides duration were removed. This approach resulted in identifying 117 to 199 strides (mean 158.6) from the 3 min walking trial, of which between 66 to 103 (83.7) were matched within 0.2 s with force plate-measured stride initiations, and the results of these matched strides are reported.

The gradient descent algorithm approach developed by Madwick and colleagues (2011) [24] was used to calculate the vertical acceleration from the waist-worn IMU-sampled accelerations and gyrations. Heel-strikes were identified by convolving the vertical acceleration with a 16 Hz Ricker wavelet (except for one participant where an 8 Hz Ricker wavelet was used instead) and then low-pass filtering the convolved signal with a 4th order zero-lag 6 Hz low-pass Butterworth filter (Figure 1). Local maxima with a prominence more than 0.5 times the 80th percentile of all the local maxima values were then identified and used to indicate heel-strikes. The heel-strikes that fell within 0.2 s of the force plate-based heel-strikes were considered as potential strides. The stride duration was defined as the difference between the current and current + 2 heel-strike. The potential stride durations were examined and those with stride duration within 1.25 of the median were included and are reported. The approach resulted in 222 to 332 steps (mean 290.0) identified from the 3 min walking trial, of which between 54 and 97 strides (mean 78.5) were matched to the force plate and included in the final sample.

We followed the approach reported by Ihlen and colleagues. The gait bouts from each day were divided into non-overlapping 1 min epochs, with the remainder not constituting a full 1 min discarded. The resultant accelerations corresponding to the identified full minutes were then evaluated for refined composite multiscale entropy (RCME) and refined multiscale permutation entropy (RMPE) [5]. In specifics, the entropy was calculated based on mean values with coarseness scales τ = 1 to 80 (= non-overlapping means from 1 to 80 samples averaged as a pre-processing step. The entropy was then calculated for each scale independently, resulting in one entropy value per scale). The length of the template (m) for entropy evaluation was set at m = 4 for all coarseness scales, and the tolerance (R) for RCME was set at R = 0.3 times the resultant standard deviation (SD) of the bout for all coarseness scales [5]. Of note, we have deviated from the approach outlined by Ihlen and colleagues in two notable ways. Firstly, we used the bout-resultant standard deviation to calculate the tolerance as opposed to the using the standard deviation of the pooled bout samples. This was done in an attempt to allow for variations in acceleration magnitudes between individuals. Secondly, we implemented the entropy analysis in java to reduce the computational time (available from https://github.com/tjrantal/javaMSE) instead of using the Matlab implementation published by Ihlen and colleagues [5]. The means over all of the epochs from a testing session (either 2 or 3 one minute epochs per session were identified) are reported as the outcomes. That is, 80 results each for RMPE, and for RCME, one result per each of the 80 coarseness scale.

Means (SD) are reported where applicable. As explained in the first publication from this dataset [21], the sample size of *n* = 27 was deemed appropriate based on the analysis by Glüer and colleagues [25]. The concurrent validity of the IMU-derived stride duration characteristics were evaluated based on the first measurement session by using force plate-derived characteristics as the comparison. The mean differences (bias), evaluated with a paired t-test, 95% limits of agreement (95% LoA), Pearson correlation coefficient (r) and intra-class correlation coefficient (calculated for absolute agreement, ICC), are reported to indicate validity. ICCs were used to indicate whether the agreement was poor (<0.40), fair (0.40 to <0.60), good (0.60 to 0.75) or excellent (≥0.75) [26]. Bland Altman plots were used to visualize the agreement between the methods and the reliability of repeated measures [27]. The reliability was evaluated using paired t-tests, Pearson correlation and ICC. The congruence between the force plate-derived stride duration variability and the IMU-derived multiscale entropy estimates were evaluated with a Pearson correlation. A statistical analysis was conducted using project R (version 2018-12-18 r75863, https://www.R-project.org/), and the significance level was set at *p* ≤ 0.05.

## 3. Results

The mean age, height and body masses of the participating women (*n* = 17) were 74.8 (SD 44) years, 160 (6) cm and 68.8 (9.5) kg. The corresponding values for men (*n* = 10) were 73.7 (4.1) years, 176 (7) cm and 84.2 (9.4) kg. The mean values of the stride duration and stride duration variability measured with all methods and in both measurement sessions are given in Table 1. All the methods indicated an excellent session-to-session reliability for the mean stride duration (force plate ICC = 0.95, 95% confidence interval [CI]; ankle-worn inertial measurement unit ICC = 0.95, 95% CI 0.90 to 0.98; waist-worn inertial measurement unit ICC = 0.95, 95% CI 0.91 to 0.98). The force plate indicated an excellent reliability for stride duration variability (SD ICC = 0.90, 95% CI 0.79 to 0.95; coefficient of variation ICC = 0.86, 95% CI 0.75 to 0.93), whereas the waist-worn IMU indicated a good reliability (SD ICC = 0.66, 95% CI 0.43 to 0.81; coefficient of variation ICC = 0.58, 95% CI 0.32 to 0.76) and the ankle-worn IMU indicated a fair reliability (SD ICC = 0.47, 95% CI 0.18 to 0.68; coefficient of variation ICC = 0.41, 95% CI 0.12 to 0.64) (Figure 2).

A significant difference (mean bias) was observed between the force plate and both the waist-worn and the ankle-worn IMU-derived stride durations (from −1.5 to −1.7 ms), stride duration SDs (from −1.8 to −6.8 ms) and stride duration coefficients of variation (from −0.18% to −0.66%) (all *p* < 0.001). An excellent agreement was observed with both IMU wear-locations for the mean stride duration (ICC = 1.00). For stride duration SD and the coefficient of variation, an excellent agreement was observed with the waist-worn IMU (ICC = 0.89 to 0.93), whereas a poor agreement was observed with the ankle-worn IMU (ICC = 0.10 to 0.22) (Table 2 and Table 3).

For RCME, the waist-worn IMU indicated a good to excellent reliability up to coarseness scale 63 with ICCs ranging from 0.60 to 0.82, whereas coarseness scales of 64–80 indicated a fair to good reliability (ICC = 0.45–0.66). Most coarseness scales from 35 upwards indicated significantly lower entropy (4.7%–6.9%) on the second week compared to the first. The coefficients of variations varied from 7.3% to 13.3% along the scales. RMPE indicated a good to excellent reliability (ICC = 0.60–0.80) apart from scales 54 to 55, which indicated a fair reliability (ICC = 0.58–0.59). No difference between measurement weeks was observed on any coarseness scale. The coefficients of variations ranged from 1.3% to 5.4% (see Appendix A for the RCME and RMPE values).

Correlation analyses between the waist-worn IMU multiscale entropy analyses and force plate-based stride duration variability indicated a significant positive association between the two for RCME from scale 20 upwards (except scale 64) (*r* = 0.40–0.83; *p* < 0.05) and for RMPE from scale 35 upwards (except scale 49) (*r* = 0.38 to 0.69; *p* < 0.05) (Figure 3).

## 4. Discussion

The primary findings of the present study were (1), that the waist-worn IMU multiscale sample entropy estimates exhibit a fair to excellent week-to-week reliability and reasonable congruency with the force plate-estimated stride duration variability and (2), ankle-worn IMU-based stride duration variability has a poor concurrent validity compared to the force plate-assessed stride duration variability and exhibits a fair reliability. Furthermore, the waist-worn IMU exhibited an excellent concurrent validity on the stride duration SD with the force plate and a good week-to-week reliability. Taken together, the findings indicate that the multiscale entropy estimates based on waist-worn IMU recordings provide a reasonable and reliable indication of gait variability among older ambulatory community-dwelling men and women in a laboratory setting.

The mean and variation values reported in the present study were well aligned with values reported from comparable populations. That is, a reference value of around 1100 ms has been reported for 65 to 74 year-olds [28], and the present mean fell within 60 ms of that. Similarly, the stride duration coefficient of variation fell within 0.3% of the 2.1% reference value [28]. No population-based reference values are available for RCME or RMPE, but the present results were very closely aligned with the ones reported by Ihlen and colleagues [5]. Given that the present population had a shorter stride duration and a lower stride duration variability than the reference, it is likely that the present sample was relatively healthier than the global population mean, which is typical of a sample of people opting to take part in laboratory measurements [29].

Out of the two entropy estimates—RCME and RMPE—used in the present study, RCME seems to be a more sensitive indicator of gait variability. Firstly, more RCME scales were positively associated with the force plate-measured stride duration variability, more so than the RMPE scales, with the correlation coefficients between the coarseness scales 20 to 50 consistently higher for RCME compared to RMPE. Secondly, the force plate-measured stride duration variability indicated a lower variability on the second measurement week compared to the first, which was also indicated by RCME but not by RMPE. On the other hand, overall, RMPE indicated a better week-to-week reliability than RCME. Both the RMPE and the RCME have been reported to be able to discriminate between older individuals with a history of falls (two or more falls in the preceding year) and those with no history of falls [5]. Therefore, calculating both the RMPE and RCME may be prudent until the pragmatic value of each of the entropy estimates is better established.

The ankle-worn IMU-based stride duration variability exhibited a fair reliability and poor concurrent validity compared to the respective force plate-based estimates. This was likely caused by the difficulty in identifying heel-strike events consistently based on the IMU signals. Pacini Panebianco and colleagues [30] provided a rather comprehensive review of various heel-strike event-detection algorithms, and we also tested the peak identification-based method (data not shown) on the vertical and horizontal accelerations, which resulted in similar findings to what Pacini Panebianco and colleagues reported. That is, a marginally better concurrent validity but poorer reliability. The fundamental issue seems to be the discretization of the signal, which leads to discarding large swathes of the recorded signal while simultaneously introducing discretization error. This issue is overcome by the entropy estimates, which consider all data points of the recorded signa and require no discretization or event detection. Taken together, these results indicate that entropy (or other methods that do not require event detection, such as utilizing a sliding window to extract descriptive features [31]) estimates may provide a more reasonable approach to gait variability estimation compared to event detection-based approaches. This is, at least, when waist-worn recorded IMU signals from ambulatory older adults are considered. Although there was an excellent concurrent validity to force plate-defined values with the waist-worn IMU, the analysis relied on the concurrent force plate analyses to identify strides, which removed false heel-strike detections from further analyses. This was seen in the rather poorer week-to-week reliability compared to what would have been expected based on the excellent concurrent validity findings. Considering all of the evidence, it appears that similar methodological issues related to the discretization of the signal that applied to the ankle-worn device also apply to the waist-worn device, despite the apparently excellent concurrent validity and good reliability.

The primary limitations of the present study pertain to generalizability. Firstly, only Caucasian healthy community-dwelling older adults were included. It is well established that ethnicity is associated with preferred gait speed and cadence [32] and, hence, whether the results can be generalized to other populations remains to be shown. Moreover, we did not screen for the presence of common confounders such as osteoarthritis, mild cognitive decline or Parkinson’s disease. Secondly, our exploration of the event-based gait variability estimates was far less than exhaustive, and we used only two (waist and ankle) sensor locations. A foot-worn sensor may well provide a more accurate event detection compared to the ankle-worn (distal lower leg just above the malloeli) location used in the present study. This may possibly enable an event-based variability estimate using a wearable IMU [30]. However, we did explore more than one event detection algorithm and are fairly confident that it is challenging to conceive of an event detection algorithm that matches force plate-based heel-strike event detection based on a waist-worn or ankle-worn IMU. Thirdly, we did not use a dedicated device (e.g., an instrumented walkway) to capture heel-strike events but rather utilized a force platform and developed a novel algorithm for heel-strike event detection, which was subsequently used as the “golden standard” for the concurrent criterion validity evaluation. The force platform method did produce reliable results, and the mean stride duration SD is well aligned with the reference values reported by Beauchet and colleagues [28]. Fourthly, we utilized the minimal feasible sample size of two repeated measurements and 27 participants to evaluate the reliability of the methods. Increasing the number of participants and/or the number of repeated testing sessions could have been used to narrow the confidence intervals of the reliability estimates [25]. Finally, the gait dynamics pertaining to variability may be evaluated using various techniques without discretizing the signal (e.g., [31]). We explored only multiscale entropy in the present study and only two analytical approaches. Future studies would be needed to identify the most appropriate metrics for specific applications (e.g., screening for cognitive decline versus increased falls risk). We chose to use RCME and RMPE because these analyses have been successfully applied in free-living recordings in the literature [5,22].

In conclusion, we found that multiscale entropy based on waist-worn IMU recordings provides a reasonably reliable gait variability estimate which is reasonably congruent with force plate-based stride duration variability estimates among community-dwelling ambulatory older adults. Entropy estimates may provide a preferable method of gait variability assessment compared to gait event detection-based estimates when wearable IMUs are used to record the gait.

## Figures and Tables

**Figure 1 sensors-20-02858-f001:**
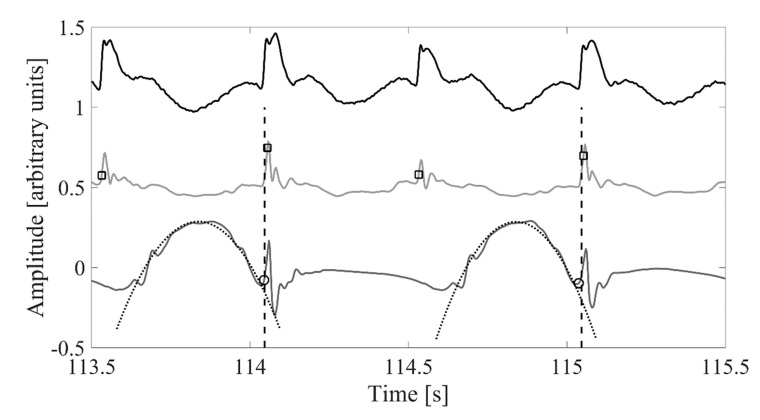
Visualization of the heel-strikes detected based on concurrently recorded ground reaction forces (vertical ground reaction—force black solid line), waist-worn inertial measurement unit (IMU) vertical acceleration (lightest grey solid line) and waist-worn IMU sagittal plane angular velocity (dark grey solid line). Dashed black vertical line indicates right foot heel-strike based on ground reaction forces. Black square indicates heel-strike based on the waist-worn IMU. Black circle indicates heel-strike based on the ankle-worn IMU recording. Dotted gray line shows the fitted parabola used to define heel-strike events based on the ankle-worn IMU recording.

**Figure 2 sensors-20-02858-f002:**
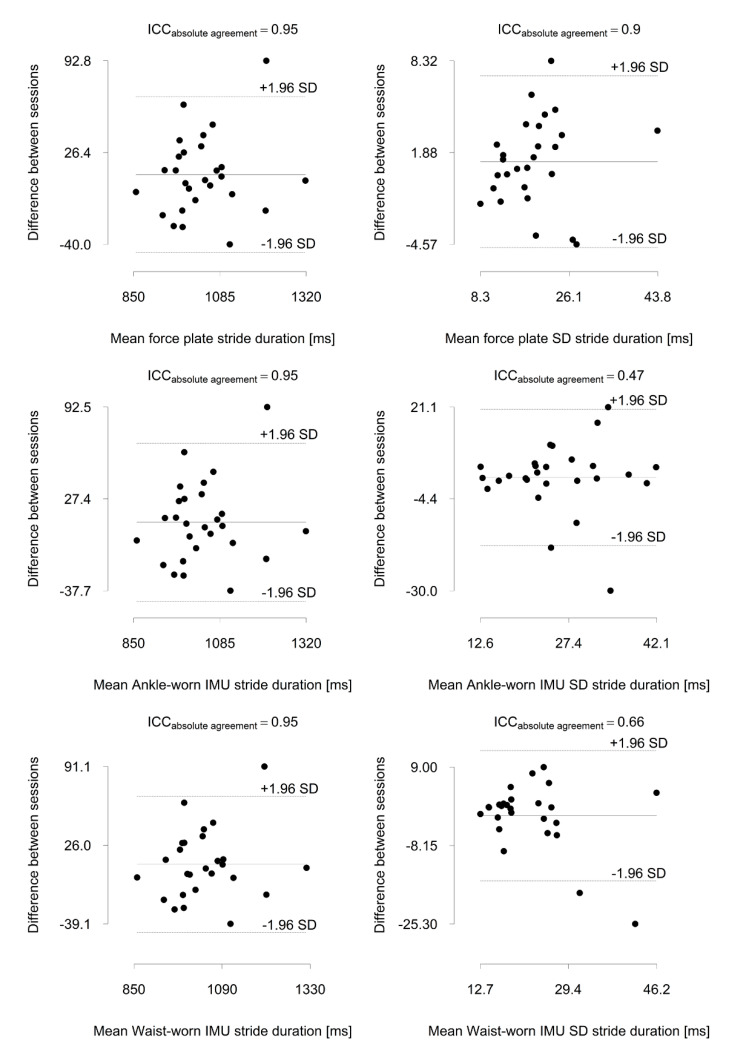
Bland-Altman plots of stride duration and stride duration variability assessed a week apart with a force plate and a wait-worn and ankle-worn inertial measurement unit (IMU). ICC = intra-class correlation coefficient.

**Figure 3 sensors-20-02858-f003:**
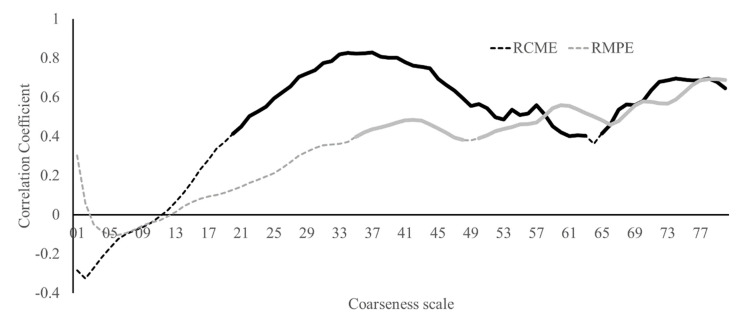
Pearson correlation coefficient calculated between the force plate-measured stride duration standard deviation and the waist-worn inertial measurement unit (IMU)-measured resultant acceleration-based multiscale sample entropy calculated with coarseness scales from 1 to 80. Solid line indicates a *p* < 0.05. The IMU sampling rate was 400 Hz, and the template length was 4 samples. RCME = refined composite multiscale entropy. RMPE = refined multiscale permutation entropy.

**Table 1 sensors-20-02858-t001:** Stride characteristics measured from *n* = 27 (waist-worn inertial measurement unit *n* = 26) participants in the first and the second measurement session a week apart with the two methods. Reliability values are given in the text.

	Stride Duration [ms]	Stride Duration SD [ms]	Stride Duration Coefficient of Variation [%]
Force plate			
Session 1	1040 (100)	19.6 (7.5)	1.86 (0.6)
Session 2	1030 (100)	18.4 (7)	1.76 (0.55)
difference (95% CI)	10 (0 to 20)	1.2 (0 to 2.5) *	0.1 (from −0.01 to 0.22)
Ankle-worn inertial measurement unit		
Session 1	1040 (100)	26.4 (9.4)	2.52 (0.87)
Session 2	1030 (100)	24.8 (9.4)	2.39 (0.83)
difference (95% CI)	10 (0 to 20)	1.5 (from −2.3 to 5.4)	0.13 (from −0.24 to 0.5)
Waist-worn inertial measurement unit		
Session 1	1050 (100)	21.5 (7.5)	2.04 (0.61)
Session 2	1040 (100)	23.1 (10)	2.23 (0.95)
difference (95% CI)	10 (0 to 20)	−1.6 (from −4.6 to 1.3)	−0.18 (from −0.48 to 0.11)

CI = confidence interval; SD = standard deviation; * *p* < 0.05.

**Table 2 sensors-20-02858-t002:** Concurrent validity of the stride characteristics measured with a force plate and an ankle-worn inertial measurement unit (session 1).

	Force Plate	Ankle-Worn Inertial Measurement Unit	Bias (95% CI)	CV%_RMS_	ICC (95% CI)	r^2^
Stride duration (ms)	1040 (100)	1040 (100)	0 (0 to 0) ***	0.1	1.00 (1.00–1.00)	1.00
Stride duration SD [ms)	19.6 (7.5)	26.4 (9.4)	−6.8 (from −10.3 to −3.3) ***	30.5	0.26 (−0.12–0.58)	0.22
Stride duration coefficient of variation [%]	1.86 (0.6)	2.52 (0.87)	−0.66 (from −1.01 to −0.31) ***	30.4	0.10 (−0.22–0.40)	0.10

CI = confidence interval; CV%_RMS_ = root-mean-squared coefficient of variation percentages. ICC = intra-class correlation coefficient calculated for absolute agreement; SD = standard deviation; *** *p* < 0.001.

**Table 3 sensors-20-02858-t003:** Concurrent validity of stride characteristics measured with a force plate and a waist-worn inertial measurement unit (session 1).

	Force Plate	Waist-Worn Inertial Measurement Unit	Bias (95% CI)	CV%_RMS_	ICC (95% CI)	r^2^
Stride duration [ms]	1040 (100)	1040 (100)	0 (0 to 0) ***	0.2	1.00 (1.00–1.00)	1.00
Stride duration SD [ms]	19.6 (7.3)	21.4 (7.4)	−1.8 (from −2.7 to −0.9) ***	11.6	0.93 (0.86–0.96)	0.91
Stride duration coefficient of variation [%]	1.86 (0.59)	2.04 (0.6)	−0.18 (from −0.27 to −0.09) ***	11.5	0.89 (0.80–0.94)	0.86

CI = confidence interval; CV%_RMS_ = root-mean-squared coefficient of variation percentage. ICC = intra-class correlation coefficient calculated for absolute agreement; SD = standard deviation; *** *p* < 0.001.

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
