# Peer review of "Gait Variability Using Waist- and Ankle-Worn Inertial Measurement Units in Healthy Older Adults"

_sensors, 2020, doi:10.3390/s20102858_

Round 1

Reviewer 1 Report

It is an interesting methodological paper, well documented and explained. It deserves to be published.

Introduction

The introduction is clear, precise with a good thread. The research question is well asked and very interesting.

I would like the authors to justify why they only use entropy. Why they did not calculate indices as Hurst did (with DFA for example). All these indices are subject to debate.

Material and methods

The methodology is well detailed and referenced and allows other authors to replicate the study.

Results

It's amazing to see the numbers in table 1, for example, being so identical. I would like a justification. It's a little strange.

The discussion supports the results

Reviewer 2 Report

The authors identified the lack of reliability or concurrent validity of wearable sensors (waist and ankle-worn IMU) and attempted to evaluate them with a force plate upon the healthy older adult walk. The study is well-structured and well-designed with the need and contribution described clearly. Here are some of my specific comments:

LINE 45: "only few studies reported quantification of gait variability measures". I am not sure what does it mean. If the studies interested in gait variability, they naturally quantified it. 

LINE 52: what do you mean by gait variability? You are comparing stride duration variability and I am not sure if this term is correct.

LINE 72: The custom-made force plate looks interesting. Could you tell us some specifications? How many plates are there? or Does it a 3-m force plate walkway instead?

LINE 156: Given a lengthy explanation of the data derivation process, it would be great if you can summarize precisely the list of parameters to be analyzed.

FIGURE 2: I am not sure rather the Bland-Altman plots are conducted correctly, while there were insufficient descriptions in-text either. From what I understand, the plots aimed to compare two kinds of measures such that a single plot shall be paired with two methods, while the difference between two measures shall be plotted against the mean of the two measures. Please check it carefully. In addition, the "agreement" conclusion derived from the Bland-Altan plot shall concern if they were "agreeable" based on the dots within the std range as well as a systematic offset and if the offset is significant. Please refer to the formal finding description of the Bland-Altman plot. 

LINE 259: While the concurrent validity judged by comparing to a well-established tets, it became ironic because your "custom-made" force plate was not the one well-established but was the comparator for validity. Please address it in the discussion.

Reviewer 3 Report

Thank you for the opportunity to review this manuscript. The authors explore the validity and reliability of gait variability and complexity measures extracted from waist- and ankle-mounted IMUs compared to force plate data (gold standard). They found that the waist-mounted IMU was an acceptable way, in terms of validity and reliability, to measure stride/step time variability and sample entropy in walking gait for older adults, whereas the ankle-mounted IMU is less reliable and valid. Furthermore, results indicate that sample entropy measures may be a better method to quantify gait variability and complexity since it uses the entire signal as opposed to stride interval measurements that require gait events with signal discretization.

I would first like to commend the authors on a very well written manuscript that was enjoyable to read. This validity and reliability study is warranted, and will be an important piece moving forward for researchers to measure gait variability and complexity in older adults using wearable technology in more ecological settings. Overall, the manuscript does not require any major revisions, but I do have some specific comments that I would like the authors to clarify and/or edit in-text:

Introduction

Lines 32-35: The opening statement is a run-on sentence and can be split into 2 sentences. First sentence can be the definition of gait variability. The second can be its clinical application to older adults.

Line 47: Some more information on multiscale sample entropy would be beneficial. Why is this a valuable measure to quantify with older adults? You speak of general gait variability and complexity above, but specifically stating if entropy has been shown to change with age, cognitive deficits and/or health outcomes would strengthen the study's objective.

Line 53: Why was >70 years-of-age chosen? If it's based on previous literature, then this should be mentioned earlier in the introduction to provide reasoning for this population.

Materials and Methods

Lines 59-60: Were there any other exclusion criteria with respect to gait function (e.g., Parkinson's, osteoarthritis)? With this age-group, there may have been other impairments. Were these recorded and may potentially be regarded as confounding variables?

Line 69-70: Placing the IMU "around the L4 level" seems fairly subjective, especially if you are assessing inter-session reliability. With a subtle shift in IMU position, this would shift the signal and potentially affect the results. Can you please comment?

Line 95-96: Coefficient of variation is also a common measure of gait variability. You will likely see similar results, but I recommend adding this measure into your analyses. 

Line 99 (Figure 1 description): It would be good to remind the reader that these are only "Right" heel strikes in the description.

Line 129: The detailed explanation by Ihlen and colleagues can be incorporated into your introduction to describe the entropy measurement (relating to my prior comment).

Line 140-141: Why did you deviate from the approach outlined by Ihlen and colleagues? A brief statement to explain this is needed in the manuscript.

148-150:  This is a nice paper to reference for your sample size. I think this could also be incorporated into the limitations section to state that more participants and/or more measurements (number of days) to increase the degrees of freedom would strengthen the reliability measurements.

Results

Table 1: The font is different for stride duration SD for waist-worn IMU.

Discussion

Lines 223-224 (and in general): Are your values of stride time, SD, and entropy on par with other papers studying a similar population? Please compare your measures to previously published data.

Lines 256-258: A sliding-window approach with acceleration signals has been used to characterize running gait along with identifying gait events (see Benson et al.'s work from Sensors (2019) and Journal of Biomechanics (2018)). This (or other methods) can be discussed that may help alleviate the troubles with signal discretization.

Line 259-260: How would the race or ethnicity of your participants affect concurrent validity and inter-session reliability?

Line 268-269: Same as above with sliding-window techniques. The examples above may be included here.
